# Multiobjective intuitionistic fuzzy programming under pessimistic and optimistic applications in multivariate stratified sample allocation problems

**Yashpal Singh Raghav**[1], **Ahteshamul Haq**[2¤]*, **Irfan Ali**[2]

**1** Department of Mathematics, Jazan University, Jizan, Saudi Arabia, **2** Department of Statistics and Operations Research, Aligarh Muslim University, Aligarh, India

¤ Current address: Department of Mathematics & Statistics, Integral University, Lucknow, India
* a.haq@myamu.ac.in

**Data Availability Statement:** All relevant data are within the paper, and additional data associated

## Abstract

This study investigates the compromise allocation of multivariate stratified sampling with complete response and nonresponse. We have formulated a multivariate stratified sampling problem as a mathematical programming problem to estimate $p$-population means with complete response and nonresponse for a fixed cost. Then, the compromise allocations for sample designs are determined by implementing intuitionistic fuzzy programming using optimistic and pessimistic solution strategies. A simulation study is carried out using the Stratify R software program to demonstrate the complete solution process. In wildlife, agricultural and marketing-related surveys, the study could be helpful. Also, the national planning policies related to surveys in such cases this study could be helpful. This study is an attempt to solve the sampling optimization problem using the Lingo-18 optimization program.

## Introduction

The sample size selected from each stratum in a stratified random sample survey (SRSS) must be known and chosen to lower the survey cost, or the estimator's sample variance must be determined to determine the estimate's precision. When the population mean of the characteristics is essential for the L strata of the stratified sampling, sample allocation $n_h^*(h = 1, 2, \ldots, L)$ helps maintain the variance of the stratified sample. In the $h^{\text{th}}$ stratum ($h = 1,2,\ldots, L$), $N_h$ denotes stratum size, $\bar{Y}_h$ stands for the population mean, $W_h = \frac{N_h}{N}$ denotes the stratum weight, and $S_h^2$ indicates the population mean square. The population mean for the $j^{\text{th}}$ character $\bar{Y}_j(j = 1, 2, \ldots, p)$ is denoted by $\bar{y}_{jst}$. Then, the sampling variance of $\bar{y}_{jst}$ is given by:

$$V(\bar{y}_{jst}) = \sum_{h=1}^{L} \frac{W_h^2 S_{hj}^2}{n_i} - \sum_{h=1}^{L} \frac{W_h^2 S_{hj}^2}{N_i}, \quad j = 1, 2, \ldots, p$$

where, $S_{hj}^2 = \frac{1}{N_h-1}\sum_{i=1}^{N_h} (y_{ihj} - \bar{Y}_{hj})^2$ is the variance for $j^{\text{th}}$ character in the $h^{\text{th}}$ stratum.

with this article can be found here (https://doi.org/
10.1007/s10852-011-9164-2).

**Funding:** The author(s) received no specific
funding for this work.

**Competing interests:** The authors have declared
that no competing interests exist.

In the case of nonresponse, let the $h^{th}$ stratum be split into $N_{h1}$, the size of the respondent's group and $N_{h2} = N - N_{h1}$, the size of the non-respondent's group. The sample size of the $h^{th}$ stratum ($n_h$ unit) is split into $n_{h1}$ units for the respondent's group and those left over for the non-respondent group with $n_{h2} = n_h - n_{h1}$ units. For the nonresponse problem, subsamples of size were obtained from the non-respondent group for the second attempt as follows

$$r_h = n_{h2}/k_h; \quad \forall k_h \geq 1, \quad h = 1, 2, \ldots, L$$

where, among the $h^{th}$ stratum of people who did not fill out the survey, the sample proportion is represented by $1/k_h$. The unbiased estimate of the $N_{h1}$ and $N_{h2}$ are the $\hat{N}_{h1} = n_{h1} N_h/n_h$ and $\hat{N}_{h2} = n_{h2} N_h/n_h$.

When utilizing the Hansen–Hurwitz method, the following formula is used to calculate the estimator of the stratum mean $\bar{Y}_{jh}$ in the $h^{th}$ stratum for the $j^{th}$ character:

$$\bar{y}_{jh(w)} = \frac{n_{h1} \bar{y}_{jh1} + n_{h2} \bar{y}_{jh2(r_h)}}{n_h}$$

for the $h^{th}$ stratum, the sample means of $n_{h1}$ units from the respondent group and $r_h$ units from the subsample from the non-respondent group are denoted by $\bar{y}_{jh1}$ and $\bar{y}_{jh2(r_h)}; j = 1, 2, \ldots, p$.

The unbiased estimate of the variance $\bar{y}_{jh(w)}$ for the $j^{th}$ characteristic is represented by:

$$V(\bar{y}_{jh(w)}) = \left(\frac{1}{n_h} - \frac{1}{N_h}\right) S_{jh}^2 + \frac{W_{h2}^2 S_{jh2}^2}{r_h} - \frac{W_{h2} S_{jh2}^2}{n_h} j = 1, 2, \ldots, p \text{ and } h = 1, 2, \ldots, L$$

and

$$S_{jh}^2 = \frac{1}{N_h - 1} \sum_{i=1}^{N_h} (y_{jhi} - \bar{Y}_{jh})^2 \text{ and } \bar{Y}_{jh} = \sum_{i=1}^{N_h} y_{jhi}/N_h$$

$$S_{jh2}^2 = \frac{1}{\hat{N}_{h2} - 1} \sum_{i=1}^{\hat{N}_{h2}} (y_{jhi} - \bar{Y}_{jh2})^2 \text{ and } \bar{Y}_{jh2} = \sum_{i=1}^{\hat{N}_{h2}} y_{jhi}/\hat{N}_{h2}$$

$W_{h2} = N_{h2}/N_h$ is the weight for the non-respondent group in the $h^{th}$ stratum.

An unbiased estimate of the population mean $\bar{Y}_j$ for the $j^{th}$ characteristics can be determined by

$$\bar{y}_{j(w)} = \sum_{h=1}^{L} W_h \bar{y}_{jh(w)} \text{ and}$$

$$V(\bar{y}_{j(w)}) = \sum_{h=1}^{L} W_h^2 V(\bar{y}_{jh(w)})$$

after that

$$V(\bar{y}_{j(w)}) = \sum_{h=1}^{L} \frac{W_h^2 (S_{jh}^2 - W_{h2} S_{jh2}^2)}{n_h} + \sum_{h=1}^{L} \frac{W_h^2 W_{h2}^2 S_{jh2}^2}{r_h} = V_j; \quad j = 1, 2, \ldots, p.$$

Because more than one characteristic is evaluated in multivariate SRSS, it is possible that the optimal allocation of resources for one characteristic will not be optimal for other characteristics. In compromise allocation, to arrive at a functional allocation that is, in some respects,

optimal for each characteristic, there is a condition that must be satisfied, known as a compromise. The estimator of sampling variance for the total population (or mean) with a fixed cost is optimized in the sampling literature by defining the units of sample sizes among strata. The allocation problem is defined as minimizing the overall survey cost given a fixed estimator precision. The sampling process entailed assessing the properties of each unit randomly selected from the sample. This is known as "multivariate or multiple response sampling." The resulting allocation is well known as the optimal allocation. Because the cost measurement is set and does not fluctuate from one stratum to the next, one way to decrease the total expense of the survey is to decrease the number of respondents included in the sample. Khan et al. [1] discussed the nonresponse case of multivariate SRSS in the presence of nonlinearity to obtain the optimal allocation and size of the subsample using the Lagrange Multiplier (LM) method. Díaz-García and Garay-Tápia [2] considered the allocation problem of SRSS of stochastic nonlinear programming (NLP) and solved it using different techniques such as the modified $\in$-model, LMs, V-model, chance constraints and E-model. Varshney et al. [3] discussed double sampling for multivariate SRSS of the multiobjective NLP problem and solved it with goal programming (GP) to obtain an optimal compromise allocation. Varshney et al. [4] described a solution procedure using GP for multiobjective integer NLP problems for nonresponse cases of SRSS and showed practical utility with a numerical illustration. Ali et al. [5] considered the multivariate SRSS of the nonlinear stochastic programming problem and used the $D_1$-distance, Chebyshev approximation method, and GP to obtain a compromise allocation. Ali and Haseen [6] provided geometric programming for multivariate SRSS for the case of nonresponse and used the first-phase solution to obtain the optimal allocation of the second phase by applying it as a role model of the primal-dual relationship theorem. Raghav et al. [7] discussed the solution procedure of the GP, value function, distance-based, and $\in$-constraint techniques for the multiobjective NLP problem of nonresponse cases with a numerical example. Varshney and Mradula [8] estimated the p-population mean for nonresponse cases and a solution procedure for lexicographic GP with numerical illustrations for practical utility. Haseen et al. [9] formulated a multivariate SRSS for the nonresponse case of the multiobjective stochastic programming problem, used the chance constraint to convert the NLP problem, and modified $\in$ – model. GP, $D_1$-distance and fuzzy programming were used to determine the Pareto optimal allocation of the expressed model. Khowaja et al. [10] described the multiobjective NLP problem in the presence of a quadratic cost function for different circumstances, namely, partial, complete, or null stratified sampling information. Ghufran *et al*. [11] discussed all integer NLP problems and used the measure of "minimizing the sum of the squares of the coefficient of variation for different characteristics." Using the fuzzy technique, the author developed a methodology to determine the optimal allocation for a multivariate SRSS. The NLP problem of the two-stage stratified Warner's randomized response model was studied by Ghufran et al. [12] to establish the best allocation in the presence of nonresponse. Gupta and Bari [13] dealt with a multivariate SRSS with uncertainties of fuzzy parabolic numbers and a fuzzy multiobjective NLP problem with a developed quadratic cost function. An α-cut was used to defuzzify the fuzzy parabolic number. Khan *et al*. [14] determined the sample sizes of the formulated multistage decision problem, where various strata were present with several characteristics and used dynamic programming approaches to obtain an integer solution. Table 1 below summarises some of the related works on multivariate stratified sampling topic.

Khowaja *et al*. [23] discussed linearizing the nonlinear objective functions of a multiobjective NLP problem and used the GP technique to solve the approximation of an integer linear programming problem. Ghufran *et al*. [25] dealt with a two-stage stratified Warner's randomized response, and the GP technique was used for the formulated multiobjective integer NLP problem with linear and quadratic cost functions. Khan et al. [18] used a dynamic

Table 1. Summary of research review.

| Authors | Nonresponse | Techniques used | | | | | Objective | | Discussion |
|---|---|---|---|---|---|---|---|---|---|
| | | LM | GP | LGP | FFGP | IF | Single | Multi | |
| Díaz García and Cortez [15] | | | | ✓ | | | | ✓ | distance-based method, ∈-Constraint method |
| Muhammad et al. [16] | | | | ✓ | | | | ✓ | Gama cost function, Extended LGP |
| Haq et al. [17] | ✓ | | | | ✓ | | | ✓ | Multi-choice, Gama cost function, Flexible fuzzy GP |
| Khan et al. [18] | | | | | | | ✓ | | Dynamic programming technique, Cochran's average allocation, Proportional allocation, C programming |
| Khan et al. [19] | | | ✓ | | | | | ✓ | Multiobjective NLP, Auxiliary information, Ratio and Regression method, GP |
| Varshney et al. [20] | | | ✓ | | | | | ✓ | Multiobjective NLP, GP |
| Khowaja et al. [21] | | ✓ | | | | | | ✓ | Multiobjective integer NLP problem, Compromise allocation, Compromise criterion |
| Gupta et al. [22] | | | ✓ | | | | | ✓ | Chance constraint, Auxiliary information, Chebyshev approximation, Fuzzy GP |
| Khowaja et al. [23] | | | | | | | | ✓ | Travel cost, Quadratic cost, multiobjective programming, E-constraint method, Distance-based method |
| Ghufran et al. [24] | | | | | | | | ✓ | Travel cost, multiobjective programming, E-constraint method, Distance-based method |
| **Discussed work** | **Both** | | | | | ✓ | | ✓ | **NLP, Optimistic method, Pessimistic method, IF** |

**LGP**-Lexicographic Goal programming.

**FFGP**- Flexible Fuzzy Goal Programming.

**IF**-Intuitionistic Fuzzy.

programming technique to determine the Pareto optimal allocation for the integer NLP problem of multivariate SRSS. Khowaja et al. [21] proposed a technique for attaining a compromise Pareto optimal allocation for multivariate SRSS.

The value function technique converts the formulated multiobjective integer NLP problem into a single-objective problem. The LMs technique was used to obtain an optimal compromise solution for continuous sample sizes. Varshney et al. [26] considered a multivariate SRSS with unknown stratum weight, and an optimal sampling was provided to estimate the unknown population means for nonresponse by utilizing double sampling for stratification. The authors used the GP method in the solution procedure development process. Varshney et al. [27] discussed multiobjective NLP problems for multivariate SRSS. The author minimized the individual estimated coefficients of variation using auxiliary information and a nonlinear cost function for various characteristics with a fixed budget to find a compromise solution using the GP technique. Varshney et al. [4] described the optimal allocation in multivariate SRSS for nonresponse, and a solution procedure was developed using the GP method for the multiobjective integer NLP problem. Ghufran et al. [28] discussed the optimum allocation problem by minimizing the variation coefficient for various characteristic estimators under the travel cost constraint for a multivariate SRSS. To obtain a compromise solution, the multiobjective NLP problem was solved using different techniques, such as distance-based, ∈-constraint, and value function methods. Dáz-Garcá and Cortéz [29] explored the allocation problem in multivariate SRSS and formulated it as a nonlinear problem of matrix optimization of integers with the constraints of a cost function and a predetermined sample size. They found that this problem could be resolved. Khan et al. [1] determined the optimal allocation and subsample size for different strata of multivariate SRSS for the nonresponse case and formulated it as an NLP problem. The LMs method was used to obtain the optimal allocation. An explicit formula was obtained for the optimal allocation and subsample sizes. Khan et al. [30]

discussed optimum allocation using various compromise criteria for a multiple-response SRSS.

Goli and Golmohammadi [31] proposed a multiobjective mathematical model to determine locations and perform distributions in a closed-loop supply chain under competitive conditions. Alike, Raghav *et al.* [32] explored a multiobjective mathematical programming problem to maximize the profit and minimize the cost function utilizing various methods for preventative system maintenance. Gupta et al. [33] discussed the stratified random sampling with minimization of variance and solved it with IF programming. Additionally, using Pareto-based algorithms, Tirkolaee *et al.* [34] discussed multiobjective optimization for a sustainable pollution-routing problem using cross-dock selection and found solutions using different optimization techniques. Similarly, Sangaiah *et al.* [35] addressed a robust mixed-integer linear programming model for LNG sales planning over a given time horizon, aiming to minimize the costs of the vendor. Besides, Goli *et al.* [36] developed a product portfolio optimization problem using a robust optimization approach and multiobjective invasive weed optimization algorithm. They used a robust optimization approach to address this issue, considering the profit margin uncertainty in real-world investment decisions.

## Methodology-optimistic and pessimistic method of intuitionistic fuzzy programming

The sampling problem is formulated as a univariate nonlinear mathematical programming problem. This is intended to satisfy all non-negativity criteria, and minimize the variance, while the cost is a constraint. Many scholars have conducted similar research to find the optimum or best solution by using the aforementioned or other related techniques.

The general single objective problem after the formulation is of the form:

$$Maximize \ (or \ Minimize) \ z = f(X)$$

$$Subject \ to \ g(X)(\leq or = or \geq)b$$

$$X \geq 0$$

The multiobjective multivariate stratified sampling (MOMSS) model where the variances form the objectives and cost works as a constraint. The MOMSS model is expressed as follows:

$$Min \ \{V_j(n)\} \ (j = 1, 2, \dots, p)$$
$$s.t. \ n \in F$$

$F$ is a feasible set and $V_j(n)$ ($j = 1,2,\dots,p$) represents the multiple objectives to be minimized. The intuitionistic fuzzy MOMSS model is expressed as

$$Min \ \{\tilde{V}_j(n)\} \ (j = 1, 2, \dots, p)$$
$$s.t. \ n \in F$$

$\tilde{V}_j(n)$ ($j = 1, 2, \dots, p$) are various objectives that need to be minimized under an intuitionistic fuzzy (IF) sense. The objective functions under intuitionistic fuzzy sense with the aspiration levels $g_j$; the positive admissible acceptances $t_j(j = 1,2,\dots,p)$, optimistic quantities $o_j(j = 1,2,\dots,p)$, and pessimistic quantities $q_j(j = 1,2,\dots,p)$ given by decision-makers (DMs),

where $o_j \leq q_j \leq t_j$. IFMOMSS is as follows:

$$\text{Find } n \text{ s.t.} \begin{cases} \text{Min } \{\tilde{V}_j(n)\} \tilde{<} g_j \ (j = 1, 2, \ldots, p) \\ \text{s.t. } n \in F \end{cases}$$

The membership functions (MFs) of the objective functions under the IF sets are as follows:

$$\mu_j(V_j(n)) = \begin{cases} 1, & V_j(n) \leq g_j \\ 1 + [(g_j - V_j(n))/t_j], & g_j < V_j(n) \leq g_j + t_j \\ 0, & V_j(n) > g_j + t_j \end{cases} \quad (1)$$

We construct the non-MF of $v_j(V_j(n))$, which must satisfy $\mu_j(V_j(n)) + v_j(V_j(n)) \leq 1$. For $V_j(n) \leq g_j$ and $\mu_j(V_j(n)) = 1$, $v_j(V_j(n)) = 0$; while for $V_j(n) \geq g_j$ and $0 \leq \mu_j(V_j(n)) < 1$, $0 \leq v_j(V_j(n)) < 1$. The non-MF of $V_j(n)$ can be defined as follows:

$$v_j(V_j(n)) = \begin{cases} 0, & V_j(n) \leq g_j + (1 - \lambda)(t_j - q_j) \\ 1 + \dfrac{V_j(n) - g_j + (1 - \lambda)(t_j - q_j)}{\lambda(o_j + t_j) + (1 - \lambda)q_j}, & g_j + (1 - \lambda)(t_j - q_j) < V_j(n) \leq g_j + (1 - \lambda)(t_j - q_j) \\ 1, & V_j(n) > g_j + (1 - \lambda)(t_j - q_j) \end{cases} \quad (2)$$

The parameter $\lambda \in [0,1]$; DM's preference information. The DM prefers positive feelings $\lambda \in [0.5,1]$, negative feelings $\lambda \in [0,0.5]$ and $\lambda = 0.5$ demonstrates that DM does not favor either positive or negative feelings. The unique forms of the non-MF were obtained for different parameter values $\lambda \in [0,1]$. For $\lambda = 1$, it is easy to see that $\mu_j(V_j(n)) = 0$, $v_j(V_j(n)) < 1$ in the interval $[g_j + t_j, g_j + t_j + o_j]$, the values of MFs will be zero while the values of non-MFs will be less than one. Yu *et al.* [37] called MF an optimistic method. The non-MF of $V_j(n)$ can then be defined as follows:

$$v_j(V_j(n)) = \begin{cases} 0, & V_j(n) \leq g_j \\ 1 + \dfrac{V_j(n) - (g_j + t_j + o_j)}{o_j + t_j}, & g_j < V_j(n) \leq g_j + t_j + o_j \\ 1, & V_j(n) > g_j + t_j + o_j \end{cases} \quad (3)$$

When $\lambda = 0$, for the interval $[g_j, g_j + t_j - q_j]$, we see that $v_j(V_j(n)) = 0$ $\mu_j(V_j(n)) < 1$, the values of MFs will be less than one while non-MFs will be zero. Then, the MF is called a pessimistic approach. Therefore, the non-MFs of $V_j(n)$ can be defined as follows:

$$v_j(V_j(n)) = \begin{cases} 0, & V_j(n) \leq g_j + t_j - q_j \\ 1 + \dfrac{V_j(n) - (g_j + t_j)}{q_j}, & g_j + t_j - q_j < V_j(n) \leq g_j + t_j \\ 1, & V_j(n) > g_j + t_j \end{cases} \quad (4)$$

The grouping of the optimistic and the pessimistic methods for $0 < \lambda < 1$ is called a mixed method (Fig 1).

The IFMONLP problem's flowchart can be seen in Fig 2.

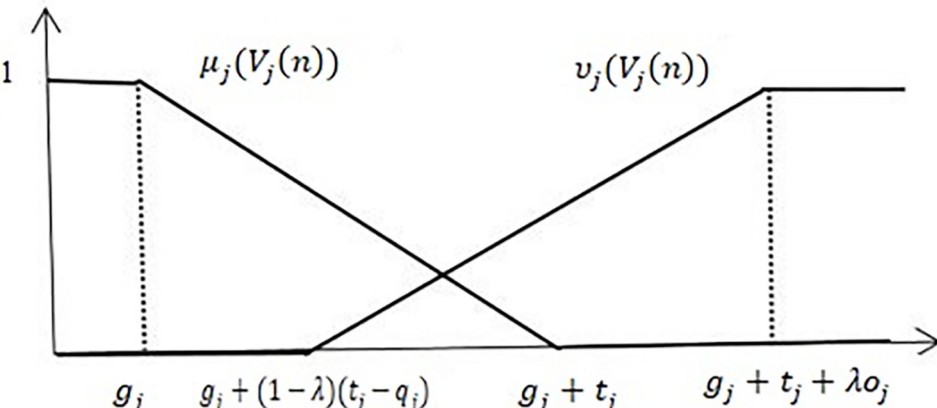

**Fig 1. MF and Non-MF.**

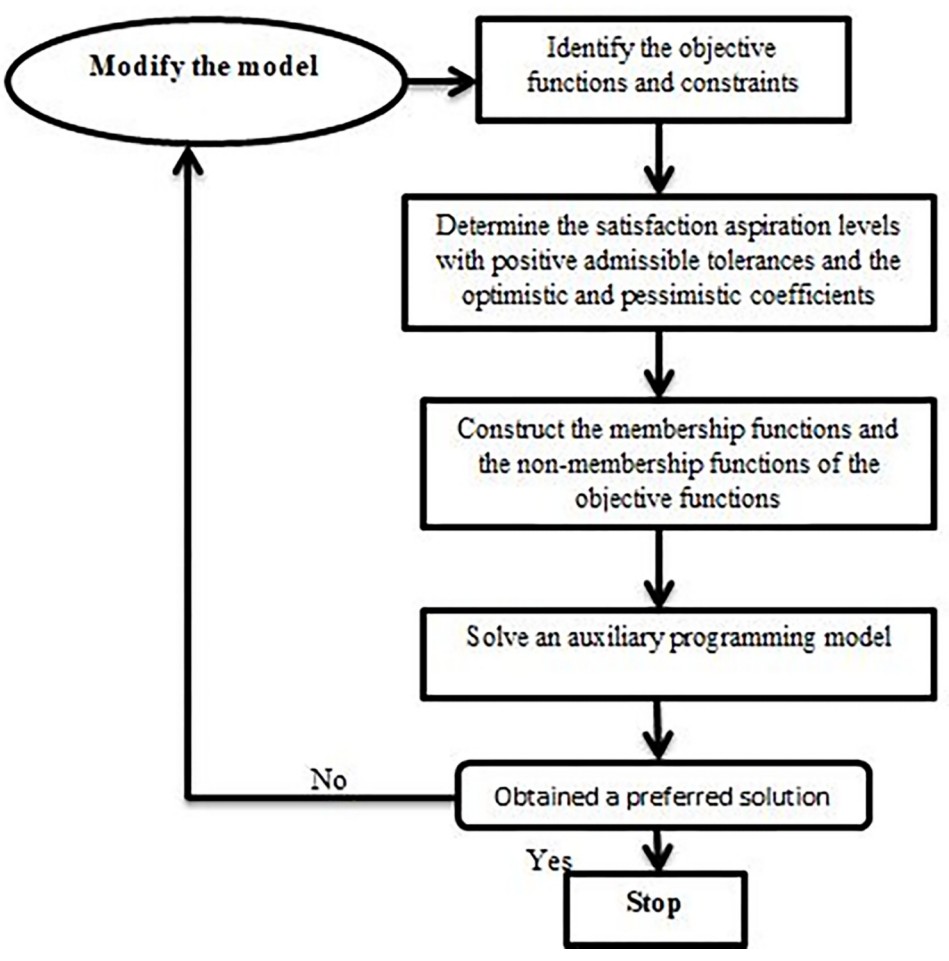

**Fig 2. The decision flowchart.**

Using Eqs 1 and 4, we solved our model as follows:

$$\text{Find } n \text{ s.t.} \begin{cases} \text{Max } (\mu - \upsilon) \\ \upsilon_j(V_j(n)) = 1 + \dfrac{V_j(n) - (g_j + t_j)}{q_j} \\ \mu_j(V_j(n)) = 1 + \dfrac{g_j - V_j(n)}{t_j} \\ \mu \leq \mu_j(V_j(n)), \upsilon \geq \upsilon_j(V_j(n)) \\ \mu_j(V_j(n)) + \upsilon_j(V_j(n)) \leq 1 \\ n \in F, \mu_j(V_j(n)) \in [0,1], \upsilon_j(V_j(n)) \in [0,1] \end{cases} \quad (5)$$

## Multivariate SRSS designs

This section discusses the multivariate stratified sampling design for optimum allocation problems under complete response and nonresponse.

### Multivariate SRSS- a case of complete response

**Defination 1:** Stratified sampling is a type of response case sampling technique in which the population is divided into subgroups (strata) based on one or more characteristics, and a sample is taken from each stratum. The goal of stratified sampling is to ensure that each stratum is represented in the sample in proportion to its representation in the population.

Several authors have discussed multiobjective optimization problems in multivariate stratified sampling subject to sampling cost and other feasibility conditions. It is also observed that the sampling costs may not always be linear; hence, the sampling costs are considered nonlinear. For instance, the cost of travel between the chosen strata units is large and cannot be ignored; it is possible that the cost function does not accurately estimate the total cost acquired. Beardwood *et al.* [38] proposed that the visiting cost $c_h$ selected for units from the $h^{\text{th}}$ stratum can be represented as $t_h\sqrt{n_h}, h = 1, 2, \ldots, L$ for the cost of travel per unit from the $h^{\text{th}}$ stratum. The distance-based estimation between $k$ randomly distributed points is related to $\sqrt{k}$. As a result of this circumstance, the sum of the costs associated with travel, measurement, and overhead will constitute the overall cost of an SRSS.

The total cost $C$ is expressed as follows:

$$C = c_0 + \sum_{h=1}^{L} c_h n_h + \sum_{h=1}^{L} t_h \sqrt{n_h}$$

Based on this discussion, the compromise allocation of a multiobjective NLP problem with a quadratic cost function is stated as follows:

$$\left. \begin{aligned} &Min V_j(n) = \sum_{h=1}^{L} \left( \frac{1}{n_h} - \frac{1}{N_h} \right) W_h^2 S_{jh}^2 \\ &\textit{subject to} \\ &\sum_{h=1}^{L} c_h n_h + \sum_{h=1}^{L} t_h \sqrt{n_h} \leq C_0; n_h \geq 0, \ h = 1, 2, \ldots, L \end{aligned} \right\}, j = 1, 2, \ldots, p$$

**Table 2. Data related to strata and their characteristics for the complete response case.**

| $h$ | $N_h$ | $S_{1h}^2$ | $S_{2h}^2$ | $S_{3h}^2$ | $t_h$ | $c_h$ |
|---|---|---|---|---|---|---|
| 1 | 395 | 35 | 1005 | 20 | 4.0 | 7 |
| 2 | 412 | 107 | 67 | 35 | 4.5 | 9 |
| 3 | 438 | 53 | 533 | 19 | 5.0 | 10 |
| 4 | 421 | 204 | 805 | 33 | 5.5 | 12 |
| 5 | 384 | 91 | 101 | 26 | 6.0 | 13 |

The available cost to meet travel and measurement expenses is $C_0 = C - c_0$; $V_j(n)$ is the sampling variance and $S_{jh}^2, h = 1, 2, \ldots, L$ are the known population variance.

**Example 1:** Data with five strata and three characteristics were simulated for a budget of 1,000 (Table 2).

Then, the compromise allocation of a multiobjective NLP problem with a quadratic cost function using the data in Table 2 is as follows:

$$\text{Minimize } V_1(n) = \frac{1.299435}{n_1} + \frac{4.321858}{n_2} + \frac{2.419448}{n_3} + \frac{8.603727}{n_4} + \frac{3.192979}{n_5}$$

$$\text{Minimize } V_2(n) = \frac{37.31234}{n_1} + \frac{2.70621}{n_2} + \frac{24.33143}{n_3} + \frac{33.95098}{n_4} + \frac{3.543856}{n_5}$$

$$\text{Minimize } V_3(n) = \frac{0.742534}{n_1} + \frac{1.413692}{n_2} + \frac{0.867349}{n_3} + \frac{1.391779}{n_4} + \frac{0.91228}{n_5}$$

$$\text{subject to } 4n_1 + 4.5n_2 + 5n_3 + 5.5n_4 + 6n_5 + 7\sqrt{n_1} + 9\sqrt{n_2} + 10\sqrt{n_3} + 12\sqrt{n_4} + 13\sqrt{n_5} \le 1000$$

$$2 \le r_h \le \hat{n}_{h2}, 2 \le n_h \le N_h$$

$\&n_h, r_h \in$ integers $\forall h = 1, 2, 3, 4, 5$.

The MFs of the IF sets that represent the objective functions are as follows:

$$\mu_1(V_1(n)) = \begin{cases} 1, & V_j(n) \le 0.6323559 \\ 1 + \dfrac{[0.6323559 - V_1(n)]}{0.4} & 0.6323559 < V_1(n) \le 1.0323559 \\ 0, & V_1(n) > 1.0323559 \end{cases}$$

$$\mu_2(V_2(n)) = \begin{cases} 1, & V_2(n) \le 2.752765 \\ 1 + \dfrac{[2.752765 - V_2(n)]}{1.29} & 2.752765 < V_2(n) \le 4.042765 \\ 0, & V_2(n) > 4.042765 \end{cases}$$

$$\mu_3(V_3(n)) = \begin{cases} 1, & V_3(n) \le 0.1802541 \\ 1 + \dfrac{[0.1802541 - V_3(n)]}{0.09} & 0.1802541 < V_3(n) \le 0.2702541 \\ 0, & V_3(n) > 0.2702541 \end{cases}$$

**Optimistic approach ($\lambda = 1$):** The non-MFs of $V_j(n) \forall j = 1,2,3$ are as follows:

$$
v_1(V_1(n)) = \begin{cases} 0, & V_1(n) \leq 0.6323559 \\ 1 + \dfrac{[V_1(n) - 1.1323559]}{0.5}, & 0.6323559 < V_1(n) \leq 1.1323559 \\ 1, & V_1(n) > 1.1323559 \end{cases}
$$

$$
v_2(V_2(n)) = \begin{cases} 0, & V_2(n) \leq 2.752765 \\ 1 + \dfrac{[V_2(n) - 5.012765]}{2.26}, & 2.752765 < V_2(n) \leq 5.012765 \\ 1, & V_2(n) > 5.012765 \end{cases}
$$

$$
v_3(V_3(n)) = \begin{cases} 0, & V_3(n) \leq 0.1802541 \\ 1 + \dfrac{[V_3(n) - 0.3102541]}{0.13}, & 0.1802541 < V_3(n) \leq 0.3102541 \\ 1, & V_3(n) > 0.3102541 \end{cases}
$$

Then, the optimistic model of the response multivariate stratified sampling is as follows:

Max $(\mu - v)$

subject to

$$\mu_1(V_1(n)) = 1 + \frac{[0.6323559 - V_1(n)]}{0.4}, \mu_2(V_2(n)) = 1 + \frac{[2.752765 - V_2(n)]}{1.29}$$

$$v_1(V_1(n)) = 1 + \frac{[V_1(n) - 1.1323559]}{0.5}, v_2(V_2(n)) = 1 + \frac{[V_2(n) - 5.012765]}{2.26}$$

$$\mu_3(V_3(n)) = 1 + \frac{[0.1802541 - V_3(n)]}{0.09}, v_3(V_3(n)) = 1 + \frac{[V_3(n) - 0.3102541]}{0.13}$$

$$V_1(n) = \frac{1.299435}{n_1} + \frac{4.321858}{n_2} + \frac{2.419448}{n_3} + \frac{8.603727}{n_4} + \frac{3.192979}{n_5}$$

$$V_2(n) = \frac{37.31234}{n_1} + \frac{2.70621}{n_2} + \frac{24.33143}{n_3} + \frac{33.95098}{n_4} + \frac{3.543856}{n_5}$$

$$V_3(n) = \frac{0.742534}{n_1} + \frac{1.413692}{n_2} + \frac{0.867349}{n_3} + \frac{1.391779}{n_4} + \frac{0.91228}{n_5}$$

$$4n_1 + 4.5n_2 + 5n_3 + 5.5n_4 + 6n_5 + 7\sqrt{n_1} + 9\sqrt{n_2} + 10\sqrt{n_3} + 12\sqrt{n_4} + 13\sqrt{n_5} \leq 1000$$

$$\mu \leq \mu_j(V_j(n)), v \geq v_j(V_j(n)), \mu_j(V_j(n)) + v_j(V_j(n)) \leq 1$$

$$\mu_j(V_j(n)) \in [0,1], v_j(V_j(n)) \in [0,1], \forall j = 1,2,3;$$

$$2 \leq n_h \leq N_h, 2 \leq r_h \leq \hat{n}_{h2}; n_h, r_h \text{ are integers}; \ h = 1,2,3,4,5.$$

We can obtain a compromise result for the integer NLP problem by utilizing the Optimistic Approach of Intuitionistic Programming. As

$$n_1 = 38, n_2 = 23, n_3 = 29, n_4 = 40, n_5 = 18, \mu = 0.8205657, v = 0.1313135$$

$$V_1 = 0.6980127, V_2 = 2.984235, V_3 = 0.1963905, \text{ Required Cost} = 999.714, \text{ Trace} = 3.878638$$

**Pessimistic approach ($\lambda = 0$):** The non-MFs of $V_j(n) \forall j = 1,2,3$ are as follows:

$$
\upsilon_1(V_1(n)) = \begin{cases} 0, & V_1(n) \leq 0.7323559 \\ 1 + \dfrac{[V_1(n) - 1.0323559]}{0.3}, & 0.7323559 < V_1(n) \leq 1.0323559 \\ 1, & V_1(n) > 1.0323559 \end{cases}
$$

$$
\upsilon_2(V_2(n)) = \begin{cases} 0, & V_2(n) \leq 2.992765 \\ 1 + \dfrac{[V_2(n) - 4.042765]}{1.05}, & 2.992765 < V_2(n) \leq 4.042765 \\ 1, & V_2(n) > 4.042765 \end{cases}
$$

$$
\upsilon_3(V_3(n)) = \begin{cases} 0, & V_3(n) \leq 0.2002541 \\ 1 + \dfrac{[V_3(n) - 0.2702541]}{0.07}, & 0.2002541 < V_3(n) \leq 0.2702541 \\ 1, & V_3(n) > 0.2702541 \end{cases}
$$

Then, the pessimistic model of the response multivariate stratified sampling is as follows:

Max $(\mu - \upsilon)$

subject to

$$\mu_1(V_1(n)) = 1 + \frac{[0.6323559 - V_1(n)]}{0.4}, \mu_2(V_2(n)) = 1 + \frac{[2.752765 - V_2(n)]}{1.29}$$

$$\upsilon_1(V_1(n)) = 1 + \frac{[V_1(n) - 1.0323559]}{0.3}, \upsilon_2(V_2(n)) = 1 + \frac{[V_2(n) - 4.042765]}{1.05}$$

$$\mu_3(V_3(n)) = 1 + \frac{[0.1802541 - V_3(n)]}{0.09}, \upsilon_3(V_3(n)) = 1 + \frac{[V_3(n) - 0.2702541]}{0.07}$$

$$V_1(n) = \frac{1.299435}{n_1} + \frac{4.321858}{n_2} + \frac{2.419448}{n_3} + \frac{8.603727}{n_4} + \frac{3.192979}{n_5}$$

$$V_2(n) = \frac{37.31234}{n_1} + \frac{2.70621}{n_2} + \frac{24.33143}{n_3} + \frac{33.95098}{n_4} + \frac{3.543856}{n_5}$$

$$V_3(n) = \frac{0.742534}{n_1} + \frac{1.413692}{n_2} + \frac{0.867349}{n_3} + \frac{1.391779}{n_4} + \frac{0.91228}{n_5}$$

$$4n_1 + 4.5n_2 + 5n_3 + 5.5n_4 + 6n_5 + 7\sqrt{n_1} + 9\sqrt{n_2} + 10\sqrt{n_3} + 12\sqrt{n_4} + 13\sqrt{n_5} \leq 1000$$

$$\mu \leq \mu_j(V_j(n)), \upsilon \geq \upsilon_j(V_j(n)), \mu_j(V_j(n)) + \upsilon_j(V_j(n)) \leq 1$$

$$\mu_j(V_j(n)) \in [0,1], \upsilon_j(V_j(n)) \in [0,1], \forall j = 1,2,3;$$

$$2 \leq r_h \leq \hat{n}_{h2}, 2 \leq n_h \leq N_h; n_h, r_h \text{ are integers}; \; h = 1,2,3,4,5$$

We can devise a compromise solution for the integer NLP problem by adopting a pessimistic approach to intuitionistic programming. As

$$n_1 = 37, n_2 = 23, n_3 = 32, n_4 = 38, n_5 = 18, \mu = 0.8247996, \upsilon = 0$$

$$V_1 = 0.7024361, V_2 = 2.976788, V_3 = 0.1959460, \text{ Cost} = 999.9377, \text{ Trace value} = 3.875170$$

## Multivariate stratified sampling- a case of nonresponse

**Defination 2:** Stratified sampling is a type of nonresponse case sampling technique in which the population is divided into subgroups (strata) based on one or more characteristics, and a

sample is taken from each stratum. The goal of stratified sampling is to ensure that each stratum is represented in the sample in proportion to its representation in the population, even if some members of the population do not respond to the survey.

The sample size $n = \sum_{h=1}^{L} n_h$, from the $n_h$ units; the respondent's group belongs to $n_{h1}$ units, and the non-respondent group belongs to $n_{h2} = n_h - n_{h1}$ units. With subsamples of sizes made for nonresponse problems as:

$$r_h = n_{h2}/k_h \forall k_h \geq 1, h = 1, 2, \ldots, L.$$

It was taken from the non-respondent's collection, the sampling fraction $1/k_h$ between the $h^{\text{th}}$ stratum of the non-respondents. The unbiased estimates of $N_{h1}$ and $N_{h2}$ are $\hat{N}_{h1} = n_{h1}N_h/n_h$ and, $\hat{N}_{h2} = n_{h2}N_h/n_h$ respectively.

Similarly, the multivariate sampling problem in the case of nonresponse can be formulated as a multiobjective nonlinear multivariate stratified sampling problem, as follows:

$$\text{Minimize} \quad V_j(n) = \sum_{h=1}^{L} \frac{W_h^2(S_{jh}^2 - W_{h2}S_{jh2}^2)}{n_h} + \sum_{h=1}^{L} \frac{W_h^2 W_{h2}^2 S_{jh2}^2}{r_h} - \sum_{h=1}^{L} \frac{W_h^2 S_{jh}^2}{N_h^2}; \forall j = 1, 2, \ldots, p$$

$$\text{subject to} \quad \sum_{h=1}^{L}(c_{h0} + c_{h1}W_{h1})n_h + \sum_{h=1}^{L}c_{h2}r_h + \omega_1 \sum_{h=1}^{L}\frac{n_h}{\lambda_1} + \omega_2 \sum_{h=1}^{L}\frac{r_h}{\lambda_2} \leq B_0$$

$$2 \leq r_h \leq \hat{n}_{h2}, 2 \leq n_h \leq N_h$$

$\& n_h, r_h \in$ integers; $h = 1, 2, \ldots, L$.

**Numerical Example 2:** With a total budget of 10,000, five strata and three data characteristics were chosen from Sukhatme et al. [39] (Table 3).

Then, the compromise allocation of a multiobjective NLP problem with a quadratic cost function using the data in Table 3 is as follows:

$$\text{Minimize } V_1(n) = \frac{0.4388203}{n_1} + \frac{2.663113}{n_2} + \frac{49.60277}{n_3} + \frac{2.66616}{n_4} + \frac{9.938173}{n_5} + \frac{0.002437891}{r_1}$$

$$+ \frac{0.08937845}{r_2} + \frac{1.206554}{r_3} + \frac{0.044436}{r_4} + \frac{0.2094497}{r_5}$$

$$\text{Minimize } V_2(n) = \frac{2.047828}{n_1} + \frac{70.97196}{n_2} + \frac{25.1802}{n_3} + \frac{11.2677}{n_4} + \frac{2.736015}{n_5} + \frac{0.03413047}{r_1}$$

$$+ \frac{2.381936}{r_2} + \frac{0.6124914}{r_3} + \frac{0.187795}{r_4} + \frac{0.05766226}{r_5}$$

$$\text{Minimize } V_3(n) = \frac{1.510273}{n_1} + \frac{0.7689739}{n_2} + \frac{0.4857774}{n_3} + \frac{0.36501}{n_4} + \frac{1.561138}{n_5} + \frac{0.02517122}{r_1}$$

$$+ \frac{0.02580803}{r_2} + \frac{0.01181621}{r_3} + \frac{0.0060835}{r_4} + \frac{0.03290141}{r_5}$$

$$\text{subject to} \quad 4n_1 + 4.9n_2 + 5.9n_3 + 7.75n_4 + 8.92n_5 + 6r_1 + 7r_2 + 9r_3 + 11r_4 + 12r_5$$

$$+ 100\sum_{h=1}^{5}\frac{n_h}{4} + 100\sum_{h=1}^{5}\frac{r_h}{4} \leq 10000$$

$$2 \leq r_h \leq \hat{n}_{h2}, 2 \leq n_h \leq N_h; n_h, r_h \text{ are integers}; \ h = 1, 2, 3, 4, 5.$$

**Table 3. Data related to the strata and their characteristics for the nonresponse case.**

| $h$ | $N_h$ | $S_{1h}^2$ | $S_{2h}^2$ | $S_{3h}^2$ | $W_{h1}$ | $W_{h2}$ | $c_{h0}$ | $c_{h1}$ | $c_{h2}$ |
|---|---|---|---|---|---|---|---|---|---|
| 1 | 395 | 12 | 56 | 41.3 | 0.75 | 0.25 | 1 | 4 | 6 |
| 2 | 382 | 80 | 2132 | 23.1 | 0.65 | 0.35 | 1 | 6 | 7 |
| 3 | 439 | 1113 | 565 | 10.9 | 0.70 | 0.30 | 1 | 7 | 9 |
| 4 | 368 | 84 | 355 | 11.5 | 0.75 | 0.25 | 1 | 9 | 11 |
| 5 | 416 | 247 | 68 | 38.8 | 0.72 | 0.28 | 1 | 11 | 12 |

The MFs for the objective functions under the IF sets are as follows:

$$\mu_1(V_1(n)) = \begin{cases} 1, & V_j(n) \leq 0.8441696 \\ 1 + \dfrac{[0.8441696 - V_1(n)]}{0.8464674} & 0.8441696 < V_1(n) \leq 1.690637 \\ 0, & V_1(n) > 1.690637 \end{cases}$$

$$\mu_2(V_2(n)) = \begin{cases} 1, & V_2(n) \leq 1.663423 \\ 1 + \dfrac{[1.663423 - V_2(n)]}{1.804234} & 1.663423 < V_2(n) \leq 3.467657 \\ 0, & V_2(n) > 3.467657 \end{cases}$$

$$\mu_3(V_3(n)) = \begin{cases} 1, & V_3(n) \leq 0.0917367 \\ 1 + \dfrac{[0.0917367 - V_3(n)]}{0.1124883} & 0.0917367 < V_3(n) \leq 0.2042250 \\ 0, & V_3(n) > 0.2042250 \end{cases}$$

**Optimistic approach ($\lambda = 1$):** The non-MFs of $V_j(n) \forall j = 1,2,3$ are as follows:

$$\upsilon_1(V_1(n)) = \begin{cases} 0, & V_1(n) \leq 0.8441696 \\ 1 + \dfrac{[V_1(n) - 2.28128374]}{1.4389348}, & 0.8441696 < V_1(n) \leq 2.28128374 \\ 1, & V_1(n) > 2.28128374 \end{cases}$$

$$\upsilon_2(V_2(n)) = \begin{cases} 0, & V_2(n) \leq 1.663423 \\ 1 + \dfrac{[V_2(n) - 4.647891]}{2.984468}, & 1.663423 < V_2(n) \leq 4.647891 \\ 1, & V_2(n) > 4.647891 \end{cases}$$

$$\upsilon_3(V_3(n)) = \begin{cases} 0, & V_3(n) \leq 0.0917367 \\ 1 + \dfrac{[V_3(n) - 0.29577133]}{0.2039766}, & 0.0917367 < V_3(n) \leq 0.29577133 \\ 1, & V_3(n) > 0.29577133 \end{cases}$$

Then, the optimistic model of the nonresponse multivariate stratified sampling is as follows:

Max $(\mu - \upsilon)$

subject to

$$\mu_1(V_1(n)) = 1 + \frac{[0.8441696 - V_1(n)]}{0.8464674}, \mu_2(V_2(n)) = 1 + \frac{[1.663423 - V_2(n)]}{1.804234}$$

$$\upsilon_1(V_1(n)) = 1 + \frac{[V_1(n) - 2.28128374]}{1.4389348}, \upsilon_2(V_2(n)) = 1 + \frac{[V_2(n) - 4.647891]}{2.984468}$$

$$\mu_3(V_3(n)) = 1 + \frac{[0.0917367 - V_3(n)]}{0.1124883}, \upsilon_3(V_3(n)) = 1 + \frac{[V_3(n) - 0.29577133]}{0.2039766}$$

$$V_1(n) = \frac{0.4388203}{n_1} + \frac{2.663113}{n_2} + \frac{49.60277}{n_3} + \frac{2.66616}{n_4} + \frac{9.938173}{n_5} + \frac{0.002437891}{r_1}$$
$$+ \frac{0.08937845}{r_2} + \frac{1.206554}{r_3} + \frac{0.044436}{r_4} + \frac{0.2094497}{r_5}$$

$$V_2(n) = \frac{2.047828}{n_1} + \frac{70.97196}{n_2} + \frac{25.1802}{n_3} + \frac{11.2677}{n_4} + \frac{2.736015}{n_5} + \frac{0.03413047}{r_1}$$
$$+ \frac{2.381936}{r_2} + \frac{0.6124914}{r_3} + \frac{0.187795}{r_4} + \frac{0.05766226}{r_5}$$

$$V_3(n) = \frac{1.510273}{n_1} + \frac{0.7689739}{n_2} + \frac{0.4857774}{n_3} + \frac{0.36501}{n_4} + \frac{1.561138}{n_5} + \frac{0.02517122}{r_1}$$
$$+ \frac{0.02580803}{r_2} + \frac{0.01181621}{r_3} + \frac{0.0060835}{r_4} + \frac{0.03290141}{r_5}$$

$$4n_1 + 4.9n_2 + 5.9n_3 + 7.75n_4 + 8.92n_5 + 6r_1 + 7r_2 + 9r_3 + 11r_4 + 12r_5$$
$$+ 100\sum_{h=1}^{5} \frac{n_h}{4} + 100\sum_{h=1}^{5} \frac{r_h}{4} \le 10000$$

$$\upsilon \ge \upsilon_j(V_j(n)), \mu \le \mu_j(V_j(n)), \mu_j(V_j(n)) + \upsilon_j(V_j(n)) \le 1$$

$$\mu_j(V_j(n)) \in [0,1], \upsilon_j(V_j(n)) \in [0,1], \forall j = 1,2,3;$$

$$2 \le r_h \le \hat{n}_{h2}, 2 \le n_h \le N_h n_h, r_h \text{ are integers; } h = 1,2,3,4,5.$$

We can devise a compromise solution for integer NLP problems using the optimistic approach of intuitionistic programming.

$$n_1 = 35, n_2 = 69, n_3 = 88, n_4 = 33, n_5 = 49, r_1 = 5, r_2 = 13, r_3 = 14, r_4 = 4, r_5 = 7$$

$$V_1 = 1.032989, V_2 = 2.059496, V_3 = 0.1168203, \text{ Cost} = 9990.13, \text{ Trace value} = 3.209306$$

**Pessimistic approach ($\lambda = 0$):** The non-MFs of $V_j(n) \forall j = 1,2,3$ are as follows:

$$\upsilon_1(V_1(n)) = \begin{cases} 0, & V_1(n) \le 1.0142696 \\ 1 + \frac{[V_1(n) - 1.690637]}{0.6763674}, & 1.0142696 < V_1(n) \le 1.690637 \\ 1, & V_1(n) > 1.690637 \end{cases}$$

$$\upsilon_2(V_2(n)) = \begin{cases} 0, & V_2(n) \le 2.054523 \\ 1 + \frac{[V_2(n) - 3.467657]}{1.413134}, & 2.054523 < V_2(n) \le 3.467657 \\ 1, & V_2(n) > 3.467657 \end{cases}$$

$$v_3(V_3(n)) = \begin{cases} 0, & V_3(n) \leq 0.1010367 \\ 1 + \dfrac{[V_3(n) - 0.2042250]}{0.1031883}, & 0.1010367 < V_3(n) \leq 0.204225 \\ 1, & V_3(n) > 0.204225 \end{cases}$$

Then, the pessimistic model of the nonresponse multivariate stratified sampling is as follows:

Max $(\mu - v)$

subject to

$$\mu_1(V_1(n)) = 1 + \frac{[0.8441696 - V_1(n)]}{0.8464674}, \mu_2(V_2(n)) = 1 + \frac{[1.663423 - V_2(n)]}{1.804234}$$

$$v_1(V_1(n)) = 1 + \frac{[V_1(n) - 1.690637]}{0.6763674}, v_2(V_2(n)) = 1 + \frac{[V_2(n) - 3.467657]}{1.413134}$$

$$\mu_3(V_3(n)) = 1 + \frac{[0.0917367 - V_3(n)]}{0.1124883}, v_3(V_3(n)) = 1 + \frac{[V_3(n) - 0.2042250]}{0.1031883}$$

$$V_1(n) = \frac{0.4388203}{n_1} + \frac{2.663113}{n_2} + \frac{49.60277}{n_3} + \frac{2.66616}{n_4} + \frac{9.938173}{n_5} + \frac{0.002437891}{r_1}$$
$$+ \frac{0.08937845}{r_2} + \frac{1.206554}{r_3} + \frac{0.044436}{r_4} + \frac{0.2094497}{r_5}$$

$$V_2(n) = \frac{2.047828}{n_1} + \frac{70.97196}{n_2} + \frac{25.1802}{n_3} + \frac{11.2677}{n_4} + \frac{2.736015}{n_5} + \frac{0.03413047}{r_1}$$
$$+ \frac{2.381936}{r_2} + \frac{0.6124914}{r_3} + \frac{0.187795}{r_4} + \frac{0.05766226}{r_5}$$

$$V_3(n) = \frac{1.510273}{n_1} + \frac{0.7689739}{n_2} + \frac{0.4857774}{n_3} + \frac{0.36501}{n_4} + \frac{1.561138}{n_5} + \frac{0.02517122}{r_1}$$
$$+ \frac{0.02580803}{r_2} + \frac{0.01181621}{r_3} + \frac{0.0060835}{r_4} + \frac{0.03290141}{r_5}$$

$$4n_1 + 4.9n_2 + 5.9n_3 + 7.75n_4 + 8.92n_5 + 6r_1 + 7r_2 + 9r_3 + 11r_4 + 12r_5$$

$$+100\sum\nolimits_{h=1}^{5} \frac{n_h}{4} + 100\sum\nolimits_{h=1}^{5} \frac{r_h}{4} \leq 10000$$

$$v \geq v_j(V_j(n)), \mu \leq \mu_j(V_j(n)), \mu_j(V_j(n)) + v_j(V_j(n)) \leq 1$$

$$v_j(V_j(n)) \in [0, 1], \mu_j(V_j(n)) \in [0, 1], \forall j = 1, 2, 3;$$

$$2 \leq r_h \leq \hat{n}_{h2}, 2 \leq n_h \leq N_h$$

$\&n_h, r_h \in$ integers $\forall h = 1, 2, 3, 4, 5.$

We can devise a compromise solution for the integer NLP problem by adopting a pessimistic approach to intuitionistic programming.

$$n_1 = 44, n_2 = 66, n_3 = 86, n_4 = 33, n_5 = 49, r_1 = 5, r_2 = 12, r_3 = 12, r_4 = 4, r_5 = 7$$

$$V_1 = 1.060225, V_2 = 2.123498, V_3 = 0.1089356, \text{Cost} = 9999.63, \text{Trace value} = 3.292658$$

## Conclusion

This paper describes the integer NLP problem discussed by the optimistic and pessimistic methods of intuitionistic fuzzy programming and solved by Lingo 18 software. Tables 4 and 5 compare the practical utility of the proposed technique.

**Table 4. Allocation of trace value with incurred survey cost for complete response case.**

| | Allocations | Trace value | Required cost |
|---|---|---|---|
| **Cochran's allocation** | $n_1 = 32, n_2 = 27, n_3 = 29, n_4 = 39, n_5 = 21$ | 3.98591 | 1009.728 |
| **Optimistic approach** | $n_1 = 38, n_2 = 23, n_3 = 29, n_4 = 40, n_5 = 18$ | 3.878638 | **999.714** |
| **Pessimistic approach** | $n_1 = 37, n_2 = 23, n_3 = 32, n_4 = 38, n_5 = 18$ | **3.875170** | 999.9377 |

**Table 5. Allocation of trace values with incurred survey cost for the nonresponse case.**

| | Allocations | Trace value | Required cost |
|---|---|---|---|
| **Cochran's allocation** | $n_1 = 37, n_2 = 69, n_3 = 82, n_4 = 37, n_5 = 51,$ $r_1 = 5, r_2 = 12, r_3 = 12, r_4 = 4, r_5 = 7$ | 3.246262 | 9961.57 |
| **Optimistic approach** | $n_1 = 35, n_2 = 69, n_3 = 88, n_4 = 33, n_5 = 49,$ $r_1 = 5, r_2 = 13, r_3 = 14, r_4 = 4, r_5 = 7$ | **3.209306** | **9990.13** |
| **Pessimistic approach** | $n_1 = 44, n_2 = 66, n_3 = 86, n_4 = 33, n_5 = 49,$ $r_1 = 5, r_2 = 12, r_3 = 12, r_4 = 4, r_5 = 7$ | 3.292658 | 9999.63 |

Table 4 shows the trace values obtained for the different allocations with respect to the total survey cost. It is observed that the model based on the pessimistic approach determines the minimum trace value compared with the model based on the optimistic approach allocation for the response case of multivariate SRSS, and the total cost is almost exhausted for this approach. Cochran's allocation, the most commonly used allocation, is also used for comparative studies. It is observed that Cochran's allocation gives the maximum value for trace and violates the cost constraint; hence, it gives an infeasible allocation, for example, 1 in multivariate case.

Table 5 demonstrates the allocation's trace value and survey cost, with the optimistic approach model determining the minimum trace value, among others. The survey cost is fully exhausted by using allocation by a pessimistic approach for the nonresponse case of multivariate SRSS. For this example, Cochran's allocation was also calculated and used for possible comparisons. The suggested technique works better for this example than the most commonly used Cochran allocation.

Based on these examples, it can be concluded that the optimistic approach model is appropriate for determining the best compromise allocation for a multivariate SRSS.

The study could be useful in marketing surveys of any new launch product and help the company how the company can improve the better quality and attractive features of their product so that they remain in the market competition. The excellent performance and underperformance products could be identified with the help of conducting such marketing surveys. In wildlife and agricultural-related surveys, the study could be helpful. National Planning policies related to surveys in this study could also be helpful.

## Acknowledgments

We would like to express our sincere thanks to all the reviewers for their valuable comments and suggestions that significantly enhanced the quality of the article. We also thank Editor-in-Chief for his great support throughout this review and publication process.

## Author Contributions

**Conceptualization:** Ahteshamul Haq, Irfan Ali.

**Data curation:** Irfan Ali.

**Funding acquisition:** Yashpal Singh Raghav.

**Methodology:** Ahteshamul Haq, Irfan Ali.

**Project administration:** Irfan Ali.

**Resources:** Yashpal Singh Raghav.

**Software:** Ahteshamul Haq.

**Supervision:** Irfan Ali.

**Validation:** Yashpal Singh Raghav, Ahteshamul Haq, Irfan Ali.

**Visualization:** Ahteshamul Haq, Irfan Ali.

**Writing – original draft:** Ahteshamul Haq.

**Writing – review & editing:** Yashpal Singh Raghav, Irfan Ali.

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
