## [Decision Letter · Decision Letter 0]

7 Feb 2023

PONE-D-23-01575Multi-objective Intuitionistic Fuzzy Programming under Pessimistic and Optimistic Applications in Multivariate Stratified Sample Allocation ProblemsPLOS ONE

Dear Dr. Haq,

Thank you for submitting your manuscript to PLOS ONE. After careful consideration, we feel that it has merit but does not fully meet PLOS ONE’s publication criteria as it currently stands. Therefore, we invite you to submit a revised version of the manuscript that addresses the points raised during the review process.

We look forward to receiving your revised manuscript.

Kind regards,

Mazyar Ghadiri Nejad, Ph.D.

Academic Editor

PLOS ONE

Journal Requirements:

"NO authors have competing interests: The authors have declared that no competing interests exist."

Reviewers' comments:

Reviewer's Responses to Questions

**Comments to the Author**

1. Is the manuscript technically sound, and do the data support the conclusions?

Reviewer #1: Yes

Reviewer #2: Partly

2. Has the statistical analysis been performed appropriately and rigorously? 

Reviewer #1: Yes

Reviewer #2: N/A

3. Have the authors made all data underlying the findings in their manuscript fully available?

Reviewer #1: Yes

Reviewer #2: Yes

4. Is the manuscript presented in an intelligible fashion and written in standard English?

Reviewer #1: Yes

Reviewer #2: No

5. Review Comments to the Author

Reviewer #1: The authors could provide good work. However, there are some concerns to be resolved.

The abstract is loosely written. It is not as informative as expected. A standard abstract must present, without leaving any doubt, the objective of the paper precisely; source of data (which is not present in your abstract) and analytical approach used; key findings and any policy implication and recommendations

The structure of the introduction is not in a standard form. first, we need to find the basics of your problems and then the main aspects and issues related to them. Do not pay attention to lots of details (especially the foundation of the problem)

English presentation of the paper should be modified so as to be more readable.

I suggest the authors read and consider the studies performed by Alireza Goli et al. and their groups on the application of metaheuristic algorithms and supply chain optimization.

Authors should improve the material selection criteria and methods section considering the latest published work in reputed journal.

The implications of the results should be discussed in more detail. The authors should provide managerial insights based on the output.

Reviewer #2: The paper deals with the challenge of allocating resources for a multivariate stratified sample survey. The goal is to estimate population means for a fixed cost and this problem is formulated as a mathematical programming problem. The paper presents solution procedures using fuzzy programming optimistic and pessimistic methods.

The paper relevance and subject is interesting. However, it can be improved further.

1. The writing of the paper needs to be reconsidered. In many cases the sentences are either grammatically incorrect or simply incomplete. For instance, in the abstract L3, the word “respectively” is used incorrectly. Or in the second sentence of the introduction, I guess “When” needs to be replace with “while” and also the last part of the sentence “helps to keep the variance of the stratified sample” is incomplete.

2. What is “s” in the notation

3. I suggest make this sentence as some bullet points, providing definitions for each symbol: “The hth stratum , , and denote stratum size, the population mean, weight, and population mean square for the hth ( ) stratum”.

4. Although the definitions and formulations in the introductions are known, proper references to the original studies should be provided.

5. The literature review provide no story and background to the current work. What is it that the authors find defective and how the use of fuzzy programming will address the shortcoming in the previous studies? I suggest reorganizing the background review in a way that the reader in informed about the gap and then authors need to clearly justify their proposal and how it will add value to the field before proceeding to methodology.

6. As mentioned earlier, the methodology needs to be rewritten in a clear way. From the first sentence “The multiobjective multivariate stratified sampling (MOMSS) model where the variances form the objectives and cost work as a constraint.” Which I find difficult to grasp what the authors are trying to deliver. Later authors define aspiration level, reservation levels … which are not used in the following models. It is not also clear how the preference information is elicited from the DM? a priori? Is it possible for the DM to provide such information before having any options or knowledge on the possibilities?

Given the interesting nature of the topic but bad flow of the paper, I would suggest authors to be given a chance to make major revisions to make the paper ready for further review and consideration.

6. PLOS authors have the option to publish the peer review history of their article (what does this mean?). If published, this will include your full peer review and any attached files.

Reviewer #1: No

Reviewer #2: No

---

## [Author Response · Author response to Decision Letter 0]

11 Mar 2023

Dear Editor

We thank you for your time and efforts on our manuscript.

Further, we take this opportunity to thank the anonymous reviewers for their valuable suggestions. The suggested major revisions have been efficiently carried out in the revised version of the manuscript.

All the changes have been highlighted in yellow.

Thanking you

---

## [Decision Letter · Decision Letter 1]

10 Apr 2023

Multiobjective Intuitionistic Fuzzy Programming under Pessimistic and Optimistic Applications in Multivariate Stratified Sample Allocation Problems

PONE-D-23-01575R1

Dear Ahteshamul Hag,

We’re pleased to inform you that your manuscript has been judged scientifically suitable for publication and will be formally accepted for publication once it meets all outstanding technical requirements.

Kind regards,

Mazyar Ghadiri Nejad, Ph.D.

Academic Editor

PLOS ONE

Additional Editor Comments (optional):

Dear respected Authors,

Considering the opinion of the respected reviewers, the manuscript is accepted to be published in the respected journal of PLOS ONE. Thanks for your efforts and hope to see another submission from you to this journal.

Regards,

Mazyar Ghadiri Nejad

Reviewers' comments:

**Comments to the Author**

1. If the authors have adequately addressed your comments raised in a previous round of review and you feel that this manuscript is now acceptable for publication, you may indicate that here to bypass the “Comments to the Author” section, enter your conflict of interest statement in the “Confidential to Editor” section, and submit your "Accept" recommendation.

Reviewer #1: All comments have been addressed

Reviewer #3: All comments have been addressed

2. Is the manuscript technically sound, and do the data support the conclusions?

Reviewer #1: Yes

Reviewer #3: Yes

3. Has the statistical analysis been performed appropriately and rigorously? 

Reviewer #1: Yes

Reviewer #3: No

4. Have the authors made all data underlying the findings in their manuscript fully available?

Reviewer #1: Yes

Reviewer #3: Yes

5. Is the manuscript presented in an intelligible fashion and written in standard English?

Reviewer #1: Yes

Reviewer #3: Yes

6. Review Comments to the Author

Reviewer #1: Authors have improved the paper well based on my prevously-mentioned comments,

So my recommendation is ACCEPT

Reviewer #3: (No Response)

7. PLOS authors have the option to publish the peer review history of their article (what does this mean?). If published, this will include your full peer review and any attached files.

Reviewer #1: No

Reviewer #3: No

---

## [Editor Report · Acceptance letter]

13 Apr 2023

PONE-D-23-01575R1 

Multiobjective intuitionistic fuzzy programming under pessimistic and optimistic applications in multivariate stratified sample allocation problems 

Dear Dr. Haq:

I'm pleased to inform you that your manuscript has been deemed suitable for publication in PLOS ONE. Congratulations! Your manuscript is now with our production department. 

Kind regards, 

on behalf of

Assoc. Prof. Dr. Mazyar Ghadiri Nejad 

Academic Editor

PLOS ONE